# Referral to Slimming World in UK Stop Smoking Services (SWISSS) versus stop smoking support alone on body weight in quitters: results of a randomised controlled trial

Deborah Lycett ![ORCID],[1] Paul Aveyard ![ORCID],[2] Andrew Farmer ![ORCID],[2] Amanda Lewis,[3] Marcus Munafò[4]

[1]Faculty of Health and Life Sciences, Coventry University, Coventry, UK
[2]Nuffield Department of Primary Care Health Sciences, University of Oxford, Oxford, UK
[3]Population Health Sciences Bristol Medical School, University of Bristol, Bristol, UK
[4]School of Experimental Psychology, University of Bristol, Bristol, UK

**Correspondence to**
Prof. Deborah Lycett;
deborah.lycett@coventry.ac.uk

## ABSTRACT

**Introduction** Most people who stop smoking gain weight. Dietary modification may seem an obvious solution, but food restriction may increase cigarette craving and smoking relapse.

**Trial design** An unblinded parallel randomised controlled trial.

**Methods** Participants were adult smokers with a body mass index greater or equal to 23 kg/m². Setting was National Health Service commissioned Stop Smoking Services, interventions were referral to a commercial weight management programme, plus stop smoking support (treatment group), compared with stop smoking support alone (control group). Objective was to compare weight change between interventions in smoking abstainers and not abstinent rates in all. Primary outcome was change in weight (kg) at 12 weeks. Randomisation sequence was computer generated and concealed until allocation.

**Results** Seventy-six participants were recruited, 37 were randomised to the treatment group and 39 to the control group. Change in weight was analysed in long-term abstainers (13 treatment, 14 control) only because the aim was to prevent weight gain associated with smoking cessation. Abstinence was analysed on an intention-to-treat basis (37 treatment, 39 control). At 12 weeks weight gain was less in the treatment than the control group with an adjusted mean difference of −2.3 kg 95% CI (−4.4 to -0.1). Craving scores were lower (Mood and Physical Symptoms Scale craving domain −1.6 (−2.7 to −0.5)) and quit rates were higher in the treatment than the control group (32% vs 21%), although the trial was not powered to superiority in cravings and quit rates. No adverse events or side effects were reported.

**Conclusion** In people who are obese and want to quit smoking, these data provide modest encouragement that providing weight management at the time of quitting may be helpful. Those who are not obese, but who are informed about potential weight gain during their quit attempt, were uninterested in a weight management programme.

**Trial registration number** ISRCTN65705512

## Strengths and limitations of this study

► The first study investigating referral to a commercial weight loss programme during smoking cessation.
► Computer generated randomisation with allocation concealment.
► Weight measured and abstinence verified by exhaled CO.
► Small study with poor recruitment but reverse power calculation showed greater than 80% power to detect change in primary outcome of weight.

## INTRODUCTION

Weight gain is a well-known consequence of smoking cessation.[1 2] A meta-analysis of weight data, from randomised controlled trials (RCTs), at 1-year postquit date showed that in untreated quitters there is a mean increase of 4–5 kg. However, variation in weight change is large, with about 16% of quitters losing weight and 13% gaining more than 10 kg.[3] Nonetheless with the majority of people gaining weight smokers may be deterred from attempting to quit and this weight gain may offset some advantages of giving up smoking. Smoking cessation-related weight gain partly explains the finding that the incidence of type 2 diabetes is increased by up to 73% in the years after cessation,[4–6] there is a 30% increased risk of hypertension,[7] and a deterioration in glycaemic control in those with diabetes[8] who quit compared with those who continue to smoke. Despite these concerns weight gain in quitters does not negate the risk reduction of cardiovascular morbidity and mortality in those successfully quit smoking.[9] There is some evidence to suggest the benefit may be mitigated[10] but the number of incident cases in higher weight gain categories are too small to draw

conclusions. Nonetheless, many people find this weight gain unacceptable and the greatest danger is it may deter individuals from quitting.[11] Yet, restricting food intake may limit successful quitting as hunger increases urges to smoke[12 13] and people who gain most weight quit more successfully.[14]

An individually tailored plan to reduce energy intake and increase exercise, with regular monitoring and adaptation of individual goals, appears to be the most promising way to prevent cessation related weight gain with the least risk of impeding a quit attempt.[15] Commercial weight management providers (CWMPs) offer this type of support and are an effective way for many to lose weight through referral from National Health Service (NHS) primary care.[16] Therefore, referral to a CWMP on prescription from the NHS Stop Smoking Services may be an effective way to prevent smoking-related weight gain without a detrimental effect on quitting success.

This study compares referral to a CWMP as part of stop smoking support with stop smoking support alone on changes in body weight in quitting smokers.

## METHODS
### Design
This was a parallel group, individually RCT to compare standard stop smoking behavioural support with an intervention that, in addition to providing standard stop smoking support, included personalised weight management support, provided by Slimming World.

This was an open label trial, as blinding patients and smoking cessation advisors to allocation to intervention or control was impossible. The primary outcome was objectively measured weight and therefore the scope for bias is limited. Stop smoking advisors, who weighed participants, did not provide the weight control support and were unlikely to have a vested interest in interpreting weight change favourably. Full details of the methods for this trial can be found in the protocol paper[17] but are summarised below.

### Setting
NHS commissioned Stop Smoking Services were first recruited from Bristol and North Somerset Primary Care Trusts. Stop Smoking Services were invited to take part by letter and a follow-up phone call 2 weeks later. Clinics were only eligible to participate if they had two new service users coming through their doors each week. Due to poor recruitment we invited more clinics (n=30) to take part in the study by widening our area further across the West Midlands and South-West England. Additionally we extended the planned duration of recruitment from 10 to 17 months, which was as far as our budgetary restraints would allow.

All individuals attending the Stop Smoking Services were invited to take part in the trial at their first visit when they set their quit date. At the second visit those wanting to participate in the trial gave their consent and were randomised to the control or intervention group. Those randomised to the intervention were given a referral to Slimming World, a location and a start date as near to their quit date. Those who declined to take part were invited to provide a reason.

Baseline measures were taken after randomisation and included age, gender, socioeconomic status, ethnicity, religion, Fagerström Test for Nicotine Dependence (FTND)[18] and all dimensions of EQ-5D (EuroQol, Rotterdam, Netherlands) as a measure of health-related quality of life[19] presented as index values.

### Participants
Daily adult smokers with expired CO >10 ppm and a body mass index (BMI) greater or equal to 23 kg/m² were included. Pregnant smokers, those with a BMI <23 kg/m², those currently losing weight, or where weight loss was contraindicated, were excluded.

Participants with an ideal BMI, as well as those who are overweight, were included in the trial. This is because help is needed not only for quitting smokers who are overweight, but also for those of a healthy weight to prevent the weight gain that is associated with smoking cessation. Without engaging in weight management strategies to counteract the physiological and behavioural responses to smoking cessation, 85% of people who quit smoking will gain weight, regardless of their starting BMI.[20]

### Randomisation
A randomisation sequence was generated using computer software. Stratified randomisation by stop smoking advisor with blocking within each stratum was used to ensure balance. The blocks were randomly ordered blocks of 2, 4 and 6. Participants were randomised 1:1 to usual care or Slimming World with usual care. Stop smoking advisors were unaware of the randomisation sequence; they opened sealed, numbered, opaque envelopes in turn.

### Interventions and comparisons
#### Trial treatment providers
Trained NHS stop smoking advisors provided participants with standard smoking cessation support in both arms of the trial. This was withdrawal oriented behavioural support focusing on key behavioural change techniques, and a prescription of nicotine replacement or varenicline to relieve withdrawal symptoms. Support was on a weekly basis beginning 2 weeks before and until 4 weeks after quit day. Weight control support was provided by trained weight management counsellors employed by Slimming World. Slimming World is a CWMP which follows the National Institute for Health and Care Excellence criteria for clinical practice. It is commissioned by the NHS to provide a weight loss service to patients. Slimming World was the CWMP of choice as it works with populations to prevent excessive weight gain rather than solely focusing on weight loss, this was pertinent to preventing cessation related weight gain.

## Intervention

In addition to usual care, participants attended Slimming World for 12 weeks receiving support to lose weight or prevent weight gain. Slimming World consultants typically agree weight management targets with their members at the first appointment and then work with their members to achieve this through regular weight monitoring, controlled amounts of high energy foods and personal eating plans.

## Control

Participants were encouraged to quit smoking first, before tackling weight. Stop smoking advisors advised satiating hunger with healthy foods, which is standard care, but did not provide detailed or personalised advice regarding weight management.

## Outcome measures
### Primary outcome measure

Change in measured weight from baseline to 12 weeks postquit day in abstinent smokers. Abstinence was prolonged abstinence, defined according to the Russell Standard.[21]

### Secondary outcome measures
► Weight at 4 weeks postquit date.
► Weight at 6 months from baseline.
► Abstinence at 4, 12 and 26 weeks.
► Participant acceptability measured quantitatively by response, attrition rates and a decliner's reasons for not participating.
► Feasibility of running the trial within the Stop Smoking Services was measured by an open text questionnaire asking for comment.

### Exploratory outcomes measures
► Change in EQ-5D index values.
► Cigarette withdrawal symptom scores as measured by the Mood and Physical Symptoms Scale (MPSS).[22]
► Associations of change in religious engagement (adapted from CSI-MEMO[23]) and religious coping (religious coping index[24]) as positive use of individual's spiritual reserve to change weight[25] and smoking behaviour[26] is increasingly evidenced.

## Statistics and data analysis
### Sample size

Difficulty in recruiting resulted in it being economically unaffordable to continue the trial; therefore, the decision was made to stop the trial early.

Following the full analysis of our outcome data we conducted reverse power calculations. This showed that we had >80% power, at an alpha error rate of 0.05, to detect the observed differences in weight change in our sample at 12 weeks between the treatment and control groups. Given a large observed difference in quit rates between the control and treatment groups we also had 90% power at an alpha error rate of 0.05 to determine non-inferiority, using a non-inferiority limit of 19%.[27]

Our original sample size had been conservative assuming we would see 20% wt. gain in the treatment group, rather than the weight loss we actually saw. This would have required 32 quitters in each trial arm at 12 weeks; instead, we achieved 8–10 quitters at that time point. To achieve 32 quitters in each trial arm, based on a conservative estimate that only 20% of those recruited would be quit at 12 weeks, we wanted to recruit 160 per trial arm. Instead, we only recruited 76 smokers in total, but the quit rates in the treatment group were higher than expected.

## Analysis

The change in weight was analysed in long-term abstainers only because only long-term abstainers gain more weight over time than the general population and because the aim was to prevent weight gain on smoking cessation.[20] Therefore, as this comparison was made in abstainers only, and not those randomised to control or treatment groups, investigation of baseline differences was conducted between these groups. To identify likely confounding variables selected baseline characteristics of age, weight, BMI, FTND, gender, ethnicity and pharmacotherapy were compared, between quitters at 4, 12 and 26 weeks postquit in the control and treatment groups, using t-tests and $\chi^2$ statistics. Weight change in the intervention and the control arm of the study was presented descriptively using means and SDs and the mean difference and 95% CIs between the two arms calculated using multiple linear regression with adjustment for baseline differences.

Abstinence was analysed on an intention-to-treat (ITT) basis, assuming all those not present at follow-up had relapsed.[21] As we recruited fewer people than intended, we could not assume effective randomisation and so investigated whether there were any differences in baseline characteristics between the control and treatment groups. Adjustment for these was then made using logistic multiple regression to obtain an ORs for becoming abstinent.

The quality of life (EQ-5D) was presented as index values calculated from interim scoring for the EQ-5D-5L spreadsheet for UK values,[28] means and SD of the EQ-5D Visual Analogue Scale (VAS) were also presented following the EQ-5D-5L reporting guide.

The change in exploratory measures were presented with descriptive statistics and analysed using multiple linear regression. We also reported descriptive data about the acceptability of the intervention, supplemented by comments from questionnaires completed by Stop Smoking Service practitioners. We analysed these using manifest content analysis of a broad surface structure reflecting what was literally said.[29] Open comments were broken down into individual codes of meaning then built up into subthemes and themes which summarised the data.

All participants gave their written informed consent to participate in the study.

## Patient and public involvement

A patient advisory group were asked to comment on the concept of the study and the perceived acceptability of the proposed intervention, feedback was positive and so study plans progressed.

## Participant reimbursement for attending follow-up visits

Participants received £5 payment for travel and inconvenience at the 12-week and 26-week follow-up visits.

## RESULTS
## Potential participants

Thirty different stop smoking clinics in the West Midlands and South-West of England inclusion criterion of treating at least two new service users attending a clinic each week. Potentially over 4400 participants could have been accessed over our 17-month recruitment period and invited to take part in the trial. Of those who declined to take part 128 individuals provided us with a reason. Eighty-three people agreed to take part and were screened for eligibility, seven were ineligible (four were already losing weight through attending a commercial weight loss provider) and 76 individuals were randomised. Of these, 37 were randomised to the treatment group and 12 returned for the 26-week follow-up, while 39 were randomised to the control group of which 10 returned for the 26-week follow-up. The majority of those not responding to follow-up did so in the first 4 weeks of the trial and continued to ignore requests thereafter. Eight withdrew from the trial for various reasons, one individual in the control group did so to join Slimming World and two individuals in the intervention group said it was too difficult to quit and diet at the same time, another reported returning to smoking (figure 1).

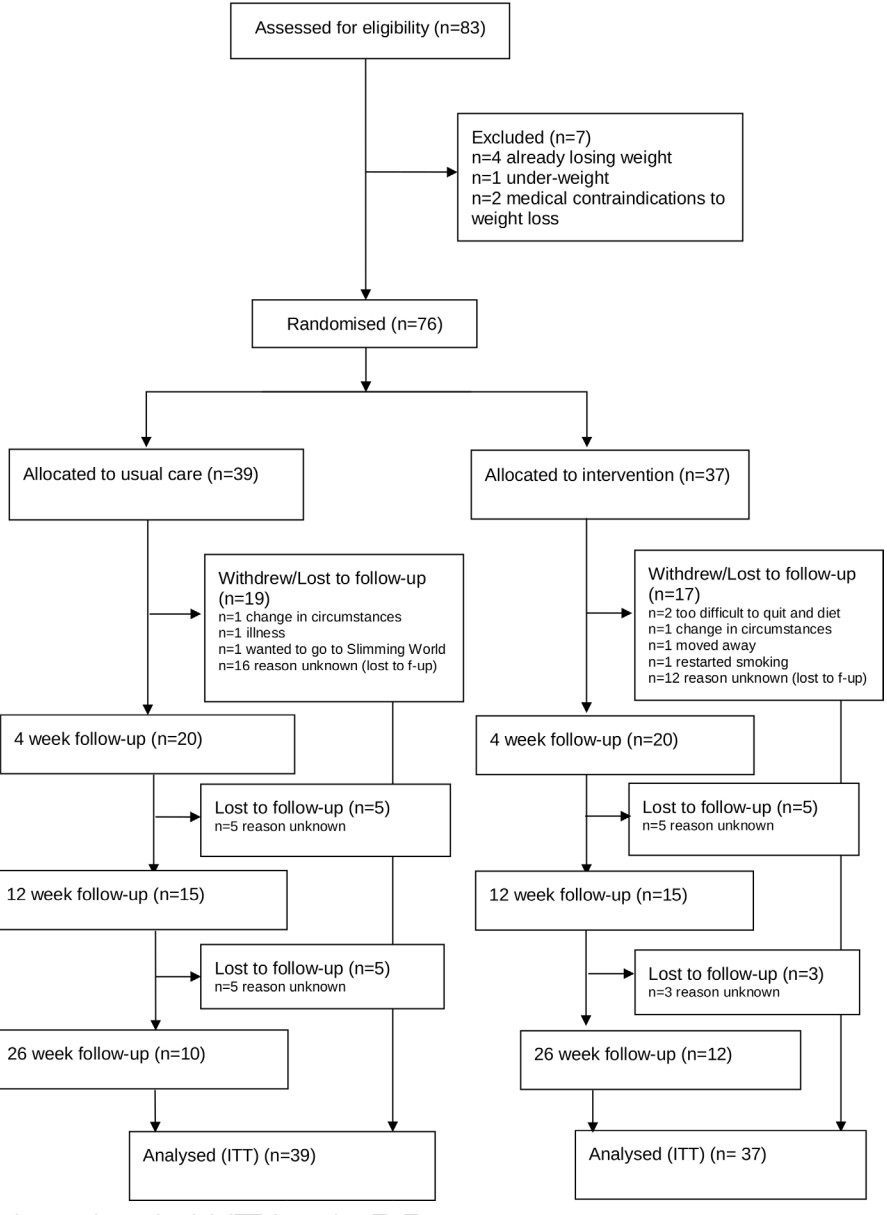

**Figure 1** Flow of participants through trial. ITT, Intention To Treat.

**Table 1** Characteristics of those who declined to take part and those who participated in the Slimming World in UK Stop Smoking Services trial

| Characteristics | Decliners (n=128) | Participants (n=76) |
|---|---|---|
| Age | 45.3 (14.1) | 46.7 (13.5) |
| Ethnicity (%) | | |
| White British | 93% (n=119) | 88% (n=67) |
| White Irish | 1% (n=1) | |
| White other | 2% (n=3) | |
| Indian | 2% (n=2) | 1% (n=1) |
| White and Asian | 3% (n=2) | |
| Black—other | 1% (n=1) | 8% (n=6) |
| Unknown | 1% (n=2) | |
| Gender | (n=71) | |
| Female | 55% (n=70) | 65% (n=46) |
| Male | 45% (n=58) | 35% (n=25) |
| BMI category | (Self-report) (n=123) | (Measured) (n=73) |
| Underweight | 7% (n=8) | 0% (n=0) |
| Healthy weight | 58% (n=71) | 12% (n=9) |
| Overweight | 35% (n=43) | 34% (n=25) |
| Obese | 1% (n=1) | 53% (n=39) |

BMI, body mass index.

### Characteristics of 'decliners' and participants

Those who provided a reason for declining to take part (decliners) and those who agreed to participate were of a similar mean age and mostly white British people. Women were more likely than men to enrol in the trial, but the most striking difference related to baseline weight status. Practically everyone who was obese agreed to join the trial, while a minority in the other BMI groups did so (table 1).

Over half those who declined to take part did so because they were not interested in weight control. (Proportionally more who gave this reason had a healthy weight rather than being overweight (data not shown).) Over 20% of reasons for declining were because people did not have time. Seventeen per cent said they were not prepared to go to Slimming World, similar numbers said they could not quit smoking and diet at the same time. Medical reasons and reasons related to the research process such as the principle of taking part or being randomised to the control were not commonly reported (online supplementary figure 1).

### Baseline characteristics of participants in the treatment and control group at baseline

Baseline characteristics of participants were investigated between the control and treatment groups as failure to recruit to target sample size could have resulted in an imbalance. Those in the control group were significantly younger and with a higher BMI than the treatment group (table 2) and adjustment was made for this in the analysis of outcomes.

### Weight change during abstinence

Among abstinent participants, there was a steady mean weight gain over time in the control group, whereas weight reduced at each time point in the treatment group. In those who were quit at 4-week follow-up there was an adjusted mean difference of −1.2 kg (−2.5 to 0.1) weight change between those in the intervention group and those in the control group. At 12 weeks the difference was greater −2.3 kg (−4.4 to −0.1) and some of this initial difference in weight loss between the trial arms was maintained at 26-week follow-up (−3.1 kg (−6.9 to 0.8)) (table 3). No confounding variables were identified in the quitters between treatment and control groups at week 4 and 12 postquit (results no shown), age was a potential confounding variable for those who remained quit at 26 weeks and adjustment for age was included in the analysis of difference in weight between the trial arms.

### Change in abstinence (intention-to-treat analysis)

In the control group 13 of 39 participants (34%) were abstinent at 4 weeks and in the treatment group 14 of 37 (38%) were abstinent. By 12 weeks 8 (21%) participants in the control and 12 (32%) in the treatment group remained abstinent. At 26-week follow-up five (13%) participants in the control group remained abstinent and eight (22%) in the treatment group.

### Change in exploratory outcomes

Change in the MPSS craving domain (MPSS-C) at all time-points showed a greater reduction in the treatment than the control group, with mean adjusted differences ranging from 0.7 to 1.9 on a scale of 0–10. In the mood domain, MPSS-M, in the longer term (26-week follow-up) there was a greater reduction in negative affect in the treatment group than the control group but the adjusted mean difference was negligible −1.3 (−3.7 to 1.2) on a scale between 7 and 35. The effect on hunger and MPSS-P within and between the trials arms was negligible at all time-points. There was negligible change in religious coping and the index values of EQ-5D between the trial arms. The VAS for quality of life improved between 4% and 6% in both groups across all time points (table 4).

### Practitioners' comments on trial delivery

Three main themes came out of the content analysis of practitioners' comments, these were 'Aspects related to research processes', which included positive aspects such as ease of implementation, however, additional time and paperwork particular on the part of the practitioner, and also the perceived burden for the participant, were viewed negatively.

Randomisation to the control group disappointed both the participant and the practitioner; practitioners reported viewing Slimming World vouchers as an incentive and this suggested misunderstanding of the nature

**Table 2** Baseline characteristics by randomisation

| Characteristic (mean (SD)/% frequency) | Control (n=39) | Treatment (n=37) |
|---|---|---|
| Age (years) | 50 (15)* | 44 (12)* |
| BMI (kg/m²) | 31.6 (6.3)* | 29.3 (4.3)* |
| FTND (scores range: 1–10) | 5.7 (2.2) | 5.8 (2.1) |
| MPSS-M (scores range: 7–35) | 12.7 (5.7) | 12.8 (4.1) |
| MPSS-C (scores range: 0–10) | 6.9 (1.7) | 7.3 (1.1) |
| MPSS-P (scores range: 3–15) | 4.2 (1.8) | 3.9 (1.1) |
| MPSS-T (scores range: 10–60) | 23.8 (7.3) | 24.1 (4.9) |
| EQ-5D | | |
| Index value | 0.759 (0.268) | 0.848 (0.105) |
| VAS (100-point scale) | 61 (23) | 67 (15) |
| Using religion to cope score (scores range: 1–10) | (n=23) | (n=21) |
| | 2.7 (2.6) | 3.7 (2.9) |
| Gender | (n=38) | (n=33) |
| Male | 29% | 44% |
| Female | 71% | 56% |
| Ethnicity | | |
| White British | 92% | 84% |
| Indian | 3% | |
| Unknown | 3% | 13% |
| Highest qualification | (n=38) | (n=32) |
| No formal qualification | 18% | 25% |
| GCSEs/O levels | 21% | 28% |
| A levels | 42% | 31% |
| Degree | 13% | 9% |
| Higher degree | 6% | 7% |
| Employment | (n=38) | (n=32) |
| Employed | 55% | 63% |
| Long-term sick | 11% | 0% |
| Looking after others | 3% | 6% |
| Retired | 21% | 13% |
| Unemployed | 8% | 13% |
| Other | 2% | 5% |
| Religious | (n=37) | (n=30) |
| No | 95% | 87% |
| Yes | 5% | 13% |
| Supportiveness of religious community | (n=20) | (n=14) |
| A lot | 5% | 35% |
| A little | 15% | 25% |
| Not at all | 80% | 30% |
| Impact of religious belief | (n=11) | (n=10) |

Continued

**Table 2** Continued

| Characteristic (mean (SD)/% frequency) | Control (n=39) | Treatment (n=37) |
|---|---|---|
| Comfort | 91% | 100% |
| Stress | 9% | 0% |
| Pharmacotherapy used | (n=16) | (n=15) |
| Nicotine replacement therapy | 38% | 40% |
| Varenicline | 62% | 60% |
| Family history of diabetes | (n=36) | (n=31) |
| Yes | 25% | 26% |
| No | 75% | 74% |
| Impaired glucose tolerance | | |
| Yes | 16% | 19% |
| No | 84% | 81% |
| Absolute diabetes prevalence | (n=32) | (n=32) |
| Yes | 14% | 13% |
| No | 86% | 87% |

*P<0.1 between control and treatment groups.

A level, advanced level exams; BMI, body mass index; FTND, Fagerström Test for Nicotine Dependence; GCSE, General Certificate of Secondary Education; O level, ordinary level exams; MPSS-C, Mood and Physical Symptoms Scale craving domain; MPSS-M, Mood and Physical Symptoms Scale mood domain; VAS, Visual Analogue Scale.

and value of randomisation and the need to promote the control arm as a good alternative. Such attitudes may have fuelled disappointment and subsequent drop out. The second theme was 'An opportunity to address weight issues', this opportunity was predominantly seen as a positive aspect of the trial and practitioners anticipated this would generate great interest from participants. However, practitioners were somewhat surprised that participants did not seem to want to address weight, there was mention

**Table 3** Change in weight in quitters at 4, 12 and 26 weeks postquit

| Change in weight (kg) from baseline in quitters at | Control | Treatment | Difference in weight change (treatment–control) mean (95% CI) adjusted for baseline weight as covariate |
|---|---|---|---|
| 4 weeks postquit | (n=13) 0.7 (1.5) | (n=14) −0.7 (1.8) | −1.2 (−2.5 to 0.1) |
| 12 weeks postquit | (n=8) 2.0 (1.9) | (n=12) −1.3 (3.3) | −2.3 (−4.4 to −0.1) |
| 26 weeks postquit | (n=5) 2.3 (2.2) | (n=8) −0.9 (2.5) | −3.1 (−6.9 to 0.8)* |

*Also adjusted for age (p<0.1 between treatment and control in 26-week quitters).

**Table 4** Change in abstinence and exploratory outcomes within and between the control and intervention group

| Change in Outcome from baseline (ITT- BOCF i.e. assumed no change) | At 4 weeks post quit | | | At 12 weeks post quit | | | At 26 weeks post quit | | |
|---|---|---|---|---|---|---|---|---|---|
| | Control | Treatment | Difference in change (treatment – control) OR / mean [95% CI]* | Control | Treatment | Difference in change (treatment – control) OR / mean [95% CI]* | Control | Treatment | Difference in change (treatment – control) % / mean [95% CI]* |
| Abstinence rate | 13/39 (34%) | 14/37 (38%) | 1.2 [0.5 to 3.1]* 1.3 [0.4 to 3.7]† | 8/39 (21%) | 12/37(32%) | 1.9 [0.7 to 5.3]* 2.9 [0.8 to 10.1]† | 5/39 (13%) | 8/37 (22%) | 1.9 [0.6 to 6.4]* 3.2 [0.7 to 14.4]2 |
| MPSS – C (range 0-10) | -0.8 (1.5) | -1.5 (2.1) | -0.7 [-1.6 to 0.2]* -0.8 [-1.8 to 0.1]† | -0.7 (1.8) | -2.2 (2.7) | -1.5 [-2.6 to -0.4]* -1.6 [-2.7 to -0.5]† | -0.7 (1.5) | -1.9 (2.5) | -1.3 [-2.2 to -0.3]* -1.5 [-2.5 to -0.4]† |
| MPSS – P (range 3-15) | 0.2 (1.3) | 0.1 (1.0) | -0.1 [-0.7 to 0.4]* <0.1 [-0.6 to 0.7]† | <0.1 (1.6) | -0.1 (1.2) | -0.1 [-0.8 to 0.5]* <0.1 [-0.7 to 0.7]† | <0.1 (1.4) | -0.1 (1.1) | 0.2 [-1.0 to 1.4]* 0.4 [-0.9 to 1.7]† |
| MPSS- M (range 7-35) | 1.1 (3.8) | 0.9 (4.9) | -0.2 [-2.4 to 1.9]* 0.1 [-2.2 to 2.4]† | -0.5 (3.7) | -1.4 (4.8) | -0.1 [-3.0 to 1.1]* -0.5 [-2.7 to 1.7]† | -0.6 (3.6) | -1.7 (5.0) | -1.3 [-3.6 to 0.9]* -1.3 [-3.7 to 1.2]† |
| Hunger (range 1-5) | 0.2 (0.8) | 0.1 (0.9) | -0.1 [-0.5 to 0.3]* <-0.1 [-0.5 to 0.4]† | 0.1 (0.6) | 0.1 (0.9) | <-0.1 [-0.4 to 0.3]* <-0.1 [-0.4 to 0.3]† | 0.1 (0.5) | 0.1 (0.9) | <-0.1 [-0.4 to 0.3]* <-0.1 [-0.4 to 0.4]† |
| MPSS- T (range 10-60) | 0.5 (4.4) | -0.6 (6.0) | -1.1 [-3.6 to 1.4]* -0.8 [-3.5 to 1.9]† | -1.3 (5.0) | -3.7 (7.4) | -2.6 [-5.6 to 0.4]* -2.1 [-5.2 to 1.1]† | -7.1 (13.4) | -7.8 (14.5) | -0.8 [-7.5 to 5.8]* -2.5 [-9.4 to 4.3]† |
| EQ-5D Index Value | 0.008(0.065) | 0.009(0.112) | 0.006 [-0.053 to 0.041]* -0.014 [-0.065 to 0.037]† | 0.0165(0.086) | 0.023(0.099) | 0.006 [-0.039 to 0.052]* 0.008 [-0.040 to 0.057]† | 0.001 (0.069) | 0.023(0.078) | 0.003 [-0.062 to 0.071]* 0.006 [-0.059 to 0.071]† |
| VAS (100 point scale) | 5.5 (20.9) | 4.4 (16.2) | -2.1 [-9.4 to 5.3]* -3.2 [-11.2 to 4.7]† | 4.8 (12.7) | 5.6 (12.7) | 0.5 [7.5 to 6.6]* -0.9 [-8.5 to 6.7]† | 5.7 (15.2) | 4.8 (11.8) | -1.1 [-7.7 to 5.5]* -2.0 [-9.0 to 5.0]† |
| Using religion to cope score (range 1-10) | -0.3 (1.2) | -0.8 (1.9) | -0.6 [-1.7 to 0.6]* -1.0 [-2.4 to 0.4]† | 0.2 (0.5) | -0.3 (0.9) | -1.3 [-2.3 to -0.4]* -1.5 [-2.8 to -0.3]† | 0.5 (1.2) | <0.1 (1.8) | -0.2 [-0.6 to 0.2]* -0.2 [-0.7 to 0.2]† |

*Unadjusted regression analysis.
†Regression analysis adjusted for baseline BMI and age.
BOCF; Baseline Observation Carried Forward; MPSS-C, Mood and Physical Symptoms Scale craving domain; MPSS-M, Mood and Physical Symptoms Scale mood domain; VAS, Visual Analogue Scale.

that fewer people at the time of this trial were seeking Stop Smoking Services than expected, and high rates of drop out were also problematic. These comments led to the third theme 'Poor participant interest and attendance' which characterised experience of delivering the trial (online supplementary table 1).

## DISCUSSION
### Summary of results and consistency with other findings
The aim of the study was to prevent cessation related weight gain—which as discussed in the introduction has both clinical and psychological consequences, the most harmful of which may be considered as relapse to smoking. The trajectory of cessation related weight gain was seen by a consistent weight increase in the control group. In comparison, the treatment group lost weight at each of these time points. Referral to the 12-week Slimming World programme plus usual stop smoking support achieved significantly less weight gain than usual stop smoking support alone. This adjusted difference (−2.3 kg (−4.4 to −0.1)) was apparent at 12 weeks postquit and is slightly greater than that seen in other trials comparing personalised weight management to usual care (−1.1 kg (−1.9 to −0.3)).[14] This was partially sustained at 26 weeks postquit (−3.2 kg (−6.9 to 0.8)).

However, we were unable to recruit to our target sample size and these results lack precision.

Percentage quit was no worse in the treatment than the control group; it was slightly higher in the treatment group but the trial was underpowered to test superiority. MPSS craving scores dropped further in the treatment than the control group (>1-point difference on 10 point scale at 12 and 26 weeks). Hunger ratings were similar in both groups across all time points.

Comparing our findings with the meta-analyses of personalised weight management versus no weight management at 26 weeks,[15] our study achieved a similar quit rate in our treatment group (22%) as in the control groups of other studies (21%). However, the quit rates in our control group was lower than in these other studies (13% vs 19%) which may account for the difference in quit rates we saw here.

Taken together these results suggest that referral to a commercial weight loss provider, such as Slimming World, at the time of a quit attempt can prevent cessation-related weight gain without negatively impacting a quit attempt. However, we cannot draw firm conclusions given our small sample size.

### Limitations
The greatest limitation to this study was the difficulty in recruiting which led to the small sample size. Anecdotally stop smoking advisors often cite weight gain is raised as a concern in clinics and were anticipating a high uptake for this trial, however, we found that when given the opportunity participation was poor and more than 50% of people

who gave a reason for declining reported it was because they were 'uninterested in weight control'.

While we do not know the reasons for everyone who declined, these results suggest that weight gain may not be as much as a barrier to quitting smoking as is commonly believed, particularly in those who are not obese. However, all those who were invited to participate in the trial had already decided to quit smoking and were looking for support in that goal. In addition, people who were already losing weight or attending a commercial weight loss programme were ineligible to take part (9% the 44 screened for eligibility). The Office for National Statistics reports weight gain only deters 5% of people from quitting and only 3% of people cite it as a reason for returning to smoking.[30] In light of this, and in view of the evidence that the cardiovascular benefits of quitting are evident despite weight gain,[9 10] we can reassure quitters who are worried about gaining weight that they need not be unduly concerned. In fact, evidence from a randomised control trial in the USA (350 quitters at 6 months) suggests that counselling quitters to reduce their weight concerns (through CBT telephone counselling compared with standard quit advice) leads to similar rates of quitting and a significant difference in weight change in favour of the counselling group (−0.5 kg vs 1.0 kg). Such counselling involved identifying and addressing maladaptive thoughts about body shape and weight, discouraged restrictive dieting, educated quitters on the relationship between weight gain and smoking cessation and encouraged an acceptance of moderate weight gain after quitting.[31]

The small sample size also means that we cannot assume effective randomisation, therefore we cannot account for any unmeasured of residual confounding which may have otherwise been explained the results we found.

The trial was also limited by a short-term follow-up at 6 months postquit (3 months post-treatment intervention), therefore the long-term consequences of the effect on weight gain prevention are unknown. However, meta-analysis data from RCTs suggests that 17% of weight lost is maintained 5 years after the end of moderate weight management interventions[32] so we might expect a similar trajectory here. This intervention in Stop Smoking Services may therefore have limited impact in the long-term prevention of weight gain associated with smoking cessation. It is also important to note that the magnitude of weight change is relatively small (mean reduction 1.1 kg at 12 weeks) only 1.3% wt. loss considering the mean weight of all participants at baseline was 86.9 kg.

Another limitation is the high rate of loss to follow-up in both the control and treatment arms of the study. The provision of free weight management advice did not deter attrition. High attrition is common in smoking cessation trials and is usually due to participants returning to smoking and as such the standard approach in analysis is to assume this.[21] However, our trial showed higher loss to follow-up compared with other smoking cessation trials,[33] this may have been due to staff time pressures

reported by practitioners which may have limited their continued attempts to make contact with those who did not attend.

Additionally, not all data were fully completed at baseline, there were disproportionally more missing values in the treatment than the control arm but we have no reason why this might be the case.

Our trial did show that a few people were interested in preventing weight gain by attending a behavioural weight loss programme, and those who wanted to take part in the trial were more likely to be female and more likely to be obese. This is consistent with other findings that suggest women who are obese are likely to be the most weight concerned.[34] Therefore, the generalisability of our findings are largely limited to these who are already overweight or obese.

For individuals who feel unable to make a quit attempt without concurrently controlling their weight, this trial suggests that referral to a behavioural weight loss programme may be a pragmatic option within stop smoking clinics.

## Implications for future trials to prevent cessation related weight gain

► Stop smoking service practitioners consider a relatively straightforward trial delivered with support to impose undue burdens on their time, limiting their willingness to approach potential participants. This is mostly due to the paperwork concerned with consent and follow-up. In low-intensity low-risk trials, we need to develop a method for achieving this that does not impose the same burdens as trials of much riskier interventions. In the meantime, trial lists need to understand this and make fewer demands by redesigning the trials.

► Most people who are stopping smoking are uninterested in active behavioural support to prevent weight gain. However, where people already are overweight to the extent of being obese, most people were prepared to enter the trial and future trials may target this population.

► Practitioners and participants quickly become despondent if not randomised to their choice of intervention, almost always the intervention group. More intensive training for practitioners in how to present trials to create equipoise is likely to be needed. There was a suggestion that quit rates were higher in the intervention than the control group and it may be helpful to explore qualitatively whether unhappiness at weight gain explains this.

## CONCLUSION

Referral to the 12-week Slimming World programme plus usual stop smoking support achieved weight loss compared with weight gain seen with usual smoking cessation support alone throughout the quit attempt as far as 26 weeks. Quit rates, cravings for cigarettes, feelings of hunger and quit rates did not seem to be

adversely affected although the recruited sample was too small to confirm this.

Recruitment into a trial of Stop Smoking Services referred into a behavioural weight loss programme was poor. They most common reason for declining to participate was no interest in controlling weight.

**Acknowledgements** We thank Carolyn Pallister and her colleagues at Slimming World for their cooperation and provision of the intervention free of charge to trial participants. We thank the Stop Smoking Services for their participation.

**Contributors** DL, PA, AF, AL and MM contributed to the design of the trial and interpretation of the data. DL conducted the analysis and drafted the manuscript. DL, PA, AF, AL and MM provided critical review on the drafts and are accountable for the work.

**Funding** This work was supported by funding of the salary for DL, provided by a fellowship from the NIHR-SPCR during the active trial period. Additional funds to set up the trial and buy equipment were provided by the NIHR-SPCR and UKCTCS. The UKCTCS is one of five UK Public Health Research Centres of Excellence. We also gratefully acknowledge funding from the British Heart Foundation, Cancer Research UK, the Economic and Social Research Council, Medical Research Council and the Department of Health, under the auspices of the UK Clinical Research Collaboration. AL's salary was paid by NIHR-SPCR during the trial period, and the salaries of AF, MM and PA were paid by their respective institutions. PA is an NIHR senior investigator and funded by NIHR Oxford Biomedical Research Centre and CLAHRC. All funding is gratefully acknowledged.

**Competing interests** DL has received hospitability from manufacturers of smoking cessation products, Pfizer, Tadworth, UK. DL's institution during the active phase of the trial has received smoking cessation products for use in a clinical trial from Johnson & Johnson, New Brunswick, New Jersey, USA. DL has received Slimming World membership vouchers for use in this trial. DL has received expenses and consultancy fees from the NHS and Universities for teaching about cessation-related weight gain. DL has received grant funding from UKCTCS and the NIHR-SPCR for research relating to cessation-related weight gain. PA is an NIHR senior investigator and is funded by NIHR Biomedical Research Centre and the CLAHRC, Oxford. AF is an NIHR Senior Investigator and receives funding from NIHR Oxford Biomedical Research Centre. AL has received hospitality from Weight Watchers. In the last 5 years, MM has received grant funding from Pfizer and varenicline, for research purposes.

**Patient consent for publication** Not required.

**Ethics approval** The study was approved by Cornwall and Plymouth NHS Research Ethics Committee. The trial was conducted in accordance with the recommendations for physicians involved in research on human subjects adopted by the 18th World Medical Assembly, Helsinki, 1964, and later revisions.

**Provenance and peer review** Not commissioned; externally peer reviewed.

**Data availability statement** Data are available upon reasonable request.

**ORCID iDs**
Deborah Lycett http://orcid.org/0000-0002-4525-6419
Paul Aveyard http://orcid.org/0000-0002-1802-4217
Andrew Farmer http://orcid.org/0000-0002-6170-4402

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
