## [Reviewer comments · BMJ Open]

ARTICLE DETAILS

TITLE (PROVISIONAL)	Referral to Slimming World in UK Stop Smoking Services (SWISS) versus Stop Smoking Support Alone on Body Weight in Quitters: Results of a Randomized Controlled Trial
AUTHORS	Lycett, Deborah; Aveyard, Paul; Farmer, Andrew; Lewis, Amanda; Munafo, Marcus

VERSION 1 – REVIEW

REVIEWER	Francesco Pistelli Cardiothoracic and Vascular Department University Hospital of Pisa Italy
REVIEW RETURNED	04-Jul-2019

GENERAL COMMENTS	General comments The present paper describes the study protocol of a open label, phase II, RCT which aims to assess whether attending a personalized weight management support (provided by Slimming World) from smoking quit date, through referral from UK NHS Stop Smoking Services, is more effective than standard smoking cessation support at preventing cessation-related weight gain (with weight change in quitters at 12 weeks post-randomization as primary outcome). The paper is well written and the research topic is of clinical and epidemiological interest for smoking cessation, with possible important implications from a public health point of view according to the results that will be observed. Specific comments Major - In the Analysis plan and/or Discussion section, it should be reported/discussed whether/how critical variables related to weight gain/control during smoking cessation will be taken into account, for example: type of pharmacotherapy used in the smoking cessation program (i.e. nicotine replacement therapy, bupropion, varenicline), baseline BMI, history of obesity, reported weight gain in previous attempts of smoking cessation, practising physical/sport activity, weight-related comorbidity.- While the present trial aims to assess whether a personalized weight management support (by Slimming World) is an effective way to prevent weight gain during smoking cessation, it seems that the participants will be left free of choosing a "modest weight loss or weight gain prevention" (page 5, left column, third paragraph), also according to whether or not they are overweight or obese at baseline. Baseline intentions of the participants with regard to lose weight or not should be recorded and taken into account in the primary analyses, since a high proportion of
---

	subjects willing to participate in the trial to loose weight might affect the outcome of the study (selection bias).  - In the Background section, more studies on the proportion of subjects who gain weight after smoking cessation should be quoted. Indeed, weight gain after smoking cessation may be frequent but it is not the rule. Minor  - Page 2, left column, lines 18-19 from the bottom. "in a recent Cochrane review update". The quoted Cochrane review is not recent, since it was published in 2012. - Table 1. The source on which the table was based (i.e. Cochrane Database Syst Rev 2012, ref no. 15) should be quoted in the legend of the table. - A reference for the EQ-5D (EuroQol, Rotterdam, Netherlands) should be provided. - The dimensions explored by the EQ-5D (EuroQol, Rotterdam, Netherlands) should be reported.
--	---

REVIEWER	Riccardo Polosa University of Catania – ITALY
REVIEW RETURNED	03-Aug-2019

GENERAL COMMENTS	The authors have compared post cessation weight change between abstainers receiving vs NOT receiving weight control support. The short-term finding of this small study (i.e. less post cessation weight gain when attending a weight management program) is anticipated. Perhaps the true strength of the paper is in the authors' analysis of the potential implications for the design of future PCWG prevention trials. I have only the following comments for improvement:  - Important confounding factors that can have impact on PCWG such as alcohol intake, being weight concern (or weight conscious) at baseline (or at time of cessation), and motivation level at keeping with regular physical activity were not considered (this becomes a critical factor for small sample size trial such as this, because distribution between study groups may be imbalanced). Please address this among the Limitations in the Discussion section. - Have all participants received the same cessation intervention? This wasn't clear – and we know Varenicline / Bupropion / NRT play different roles in attenuating PCWG. Please provide information about their distribution between study groups. - The final follow-up is only at 6 months (3 months after end of intervention). It is likely that any benefit of the weight management program will be dissipated already at 6-12 months after intervention. Ongoing intervention is obviously needed to sustain the observed beneficial effect. This aspect together with the very little interest in the weight management program by the large majority of smokers attending Stop Smoking Services should lead to a more in-depth discussion about their insignificant cost-effectiveness and lack of impact at population level. Needs to be addressed. - High rate of loss to follow-up was also reported in the intervention arm of the study, in spite of the fact that the weight management program was offered for free. This needs to be addressed. - Study findings cannot be generalized as they are restricted only to a population of obese/overweight smokers. Add to limitation of the study,
--

	- I acknowledge that prolonged abstinence is defined according to the widely used Russell Standard, but a good quality PCWG trial requires stronger measures of quantity and frequency of tobacco use and stronger compliance checks. In the specific case of eCO confirmed abstinence, the issue I have is that abstinence appears to be confirmed only at 12 and 26 weeks. Hard to establish continuous (vs intermittent) cessation with the approach described in this study. Cotinine assessments may be more useful – although if being prescribed an NRT as an intervention perhaps not possible. - The observed mean difference of -1.9 Kg at 3 months (-1.3 Kg at 6 months) between study groups is clinically (as well as psychologically) irrelevant and specifically in a population of overweight and obese smokers (88% of the total). The practical implications of such slim change must be discussed.
--	---

REVIEWER	Steve Sutton Moffitt Cancer Center and Research Institute Tampa, FL USA
REVIEW RETURNED	09-Sep-2019

GENERAL COMMENTS	The manuscript presents results for a study assessing the effectiveness of weight management treatment as a supplement to smoking cessation treatment among treatment seekers. The topic is important and the novel approach to incorporating weight management treatment is to be applauded. Unfortunately, slow enrollment eventuated in stopping the study short of the target sample size. The following are my primary concerns regarding the method. 1) The final sample is less than what was intended. It would help to provide the planned sample size and the analysis that was used to generate that target. This may be provided in the referenced protocol paper, but needs to be presented here for the reader to understand how much less the actual sample size was relative to the planned sample size. 2) A related issue is the decision to stop recruitment and end the study early. The factors that determined this decision should be described in the Introduction. This decision cannot be based on a posthoc power analysis, or justified by a posthoc power analysis. With such a small sample size for the analysis of the primary outcome, there is a greater probability that the observed group difference occurred by chance. Deciding to stop at a point in the study when the main outcome showed a significant difference may be taking advantage of chance factors. 3) Given the small sample sizes and the presented values in Table 1, I presume that there were no statistically significant group differences in the characteristics listed. It would be helpful to formally present that information in text or in the table. For any group difference with $p < .10$ (maybe higher), that characteristic should be included in any analysis focusing on treatment effects. On page 12, there is a statement that there are no treatment group differences. However, the statement directs the reader to Table 2, which presents data on 'participants' versus 'decliners'. And the statement follows a paragraph describing reasons for declining to participate. Both sets of group comparisons should be presented.
---

	4) The analysis of the primary outcome used ANCOVA. Given the small sample size and the lack of direct randomization of the data analytic treatment groups, there should be an attempt to control for baseline characteristics that may differ by analytic group and/or are associated with the outcome measure. Even marginally statistically significant differences and associations are less likely to be observed given the small sample size. Therefore, the approach may have to be conceptual. That is, the analysis would incorporate any conceptually relevant baseline characteristic, especially one that shows any sign of a difference between groups. 5) Given the small sample size, conclusions based on non-significant differences should not be highlighted (e.g., hunger ratings in the abstract). The following are relatively minor concerns. 1) There are numerous baseline measures presented in Table 1 where the number of missing observations for the Treatment group (4-7 of 37) is much greater than for the Control group (1-2 of 39). It would help to have an explanation for the degree of missing baseline data. 2) A second minor concern with Table 1 is the mean and standard deviation for FTND in the Treatment group. These values do not seem reasonable. 3) There appears to be a typographical error in Table 4. For 12 weeks and 26 weeks, the column titles for group difference in change has '%' written where I believe 'OR' was intended. 4) As I understand it, the 4-week and 26-week measures are anchored by the date of the baseline measure and the 12-week measure is anchored by quit. If there are different anchors, it would help to have descriptive statistics on the quit date relative to baseline. Table 3, though, list all follow-ups as anchored by quit date.
--	---

VERSION 1 – AUTHOR RESPONSE

Reviewer: 1

Reviewer Name

Francesco Pistelli

Institution and Country

Cardiothoracic and Vascular Department

University Hospital of Pisa

Italy

Please state any competing interests or state 'None declared':

None declared

Please leave your comments for the authors below

General comments

The present paper describes the study protocol of a open label, phase II, RCT which aims to assess whether attending a personalized weight management support (provided by Slimming World) from smoking quit date, through referral from UK NHS Stop Smoking Services, is more effective than standard smoking cessation support at preventing cessation-related weight gain (with weight change in quitters at 12 weeks post-randomization as primary outcome).

The paper is well written and the research topic is of clinical and epidemiological interest for smoking cessation, with possible important implications from a public health point of view according to the results that will be observed.

Specific comments

Major

1. - In the Analysis plan and/or Discussion section, it should be reported/discussed whether/how critical variables related to weight gain/control during smoking cessation will be taken into account, for example: type of pharmacotherapy used in the smoking cessation program (i.e. nicotine replacement therapy, bupropion, varenicline), baseline BMI, hystory of obesity, reported weight gain in previous attempts of smoking cessation, practising physical/sport activity, weight-related comorbidity.

As recruitment did not reach sample size we understand we cannot assume effective randomisation. We have now investigated whether there were any significant differences in baseline characteristics (see footnotes on Table 1) between the intervention and control groups including some of those listed above (pharmacotherapy, baseline BMI). We found a significant

difference ($p < 0.1$ as per reviewer 3's comment 3.) only for baseline BMI ($p = 0.82$) and age ($p = 0.53$) and have now added into table 4 the adjusted results for all the outcomes (adjusted for BMI and age).

We have updated the analysis section of the methods to reflect the above:

The change in weight was analysed in long-term abstainers only because only long-term abstainers gain more weight over time than the general population and because the aim was to prevent weight gain on smoking cessation [20]. **Therefore as this comparison was made in abstainers only, and not those randomised to control or treatment groups, investigation of baseline differences was conducted between these groups.** Weight change in the intervention and the control arm of the study was presented descriptively using means and SDs and the mean difference and 95% confidence intervals (CI) between the two arms calculated using **multiple linear regression with adjustment for baseline differences.**

Abstinence was analysed on an intention-to-treat basis, assuming all those not present at follow-up had relapsed [21]. As we recruited fewer people than intended, we **could not assume effective randomisation and so investigated whether there were any differences in baseline characteristics between the control and treatment groups. Adjustment for these was then made using logistic multiple regression to obtain an odds ratios for becoming abstinent.**

For those variables listed by the reviewer above, for which we do not have a data, we have added to the discussion that our conclusions are limited by the existence of unmeasured and residual confounding variables.

Added to limitations:

The small sample size also means that we cannot assume effective randomisation, therefore we cannot account for any unmeasured of residual confounding which may have otherwise been explained the results we found.

2. - While the present trial aims to assess whether a personalized weight management support (by Slimming World) is an effective way to prevent weight gain during smoking cessation, it seems that the participants will be left free of choosing a "modest weight loss or weight gain prevention" (page 5, left column, third paragraph), also according to whether or not they are overweight or obese at baseline. Baseline intentions of the participants with regard to lose weight or not should be recorded and taken into account in the primary analyses, since a high proportion of subjects willing to participate in the trial to loose weight might affect the outcome of the study (selection bias).

The comment here refers to the previously published protocol paper, we did not collect details of individual weight targets because the pragmatic intervention was individually tailored, such that targets and goals are different for each individual, according to their needs.

However we did find that those who entered the trial compared to those who declined to participate were indeed mainly those who were obese and in need of weight loss. Only 12% of participants were of a healthy weight. This was reported in the text and in Table 2 and discussed as an important consideration for the target population of future trials in this field.

Additionally the weight change outcomes were adjusted for baseline body weight as a to account for individual baseline differences.

3. - In the Background section, more studies on the proportion of subjects who gain weight after smoking cessation should be quoted. Indeed, weight gain after smoking cessation may be frequent but it is not the rule.

Added:

Weight gain is a well-known consequence of smoking cessation [1, 2]. A meta-analysis of weight data, from randomised controlled trials, at one year post quit date showed that in untreated quitters there is a mean increase of 4-5kg. However variation in weight change is large, with about 16% of quitters losing weight and 13% gaining more than 10kg [3]. Nonetheless with the majority of people gaining weight smokers may be deterred from attempting to quit and this weight gain may offset some advantages of giving up smoking.

[3] Aubin H-J, Farley A, Lycett D, Lahmek P, Aveyard P. Weight gain in smokers after quitting cigarettes: meta-analysis BMJ 2012; 345 :e4439

Minor

4. - Page 2, left column, lines 18-19 from the bottom. "in a recent Cochrane review update". The quoted Cochrane review is not recent, since it was published in 2012.

The above comment refers to the protocol paper and not this manuscript so is irrelevant.

5. - Table 1. The source on which the table was based (i.e. Cochrane Database Syst Rev 2012, ref no. 15) should be quoted in the legend of the table.

The above comment refers to the protocol paper and not this manuscript so is irrelevant.

6. - A reference for the EQ-5D (EuroQol, Rotterdam, Netherlands) should be provided.

Added reference 19:

Herdman M, Gudex C, Lloyd A, Janssen MF, Kind P, Parkin D, Bonsel G, Badia X. Development and preliminary testing of the new five-level version of EQ-5D (EQ-5D-5L). Quality of life research. 2011 Dec 1;20(10):1727-36.

7. - The dimensions explored by the EQ-5D (EuroQol, Rotterdam, Netherlands) should be reported.

Added:

...all dimensions of EQ-5D... presented as index values.

Reviewer: 2

Reviewer Name

Riccardo Polosa

Institution and Country

University of Catania - ITALY

Please state any competing interests or state 'None declared':

None declared

Please leave your comments for the authors below

The authors have compared post cessation weight change between abstainers receiving vs NOT receiving weight control support. The short-term finding of this small study (i.e. less post cessation

weight gain when attending a weight management program) is anticipated. Perhaps the true strength of the paper is in the authors' analysis of the potential implications for the design of future PCWG prevention trials. I have only the following comments for improvement:

1. - Important confounding factors that can have impact on PCWG such as alcohol intake, being weight concern (or weight conscious) at baseline (or at time of cessation), and motivation level at keeping with regular physical activity were not considered (this becomes a critical factor for small sample size trial such as this, because distribution between study groups may be imbalanced). Please address this among the Limitations in the Discussion section.

Addressed as for reviewer 1's first point.

As recruitment did not reach sample size we understand we cannot assume effective randomisation. We have now investigated whether there were any significant differences in baseline characteristics (see footnotes on Table 1) between the intervention and control groups including some of those listed above (pharmacotherapy, baseline BMI). We found a significant difference ($p < 0.1$ as per reviewer 3's comment 3.) only for baseline BMI ($p = 0.82$) and age ($p = 0.53$) have now added into table 4 the adjusted results for all the outcomes (adjusted for BMI and age).

We have updated the analysis section of the methods to reflect the above:

The change in weight was analysed in long-term abstainers only because only long-term abstainers gain more weight over time than the general population and because the aim was to prevent weight gain on smoking cessation [20]. **Therefore as this comparison was made in abstainers only, and not those randomised to control or treatment groups, investigation of baseline differences was conducted between these groups.** Weight change in the intervention and the control arm of the study was presented descriptively using means and SDs and the mean difference and 95% confidence intervals (CI) between the two arms calculated using **multiple linear regression with adjustment for baseline differences.**

Abstinence was analysed on an intention-to-treat basis, assuming all those not present at follow-up had relapsed [21]. As we recruited fewer people than intended, we **could not assume effective randomisation and so investigated whether there were any differences in baseline characteristics between the control and treatment groups. Adjustment for these was then made using logistic multiple regression to obtain an odds ratios for becoming abstinent.**

For those variables listed by the reviewer above, for which we do not have a data, we have added to the discussion that our conclusions are limited by the existence of unmeasured and residual confounding variables.

Added to limitations:

The small sample size also means that we cannot assume effective randomisation, therefore we cannot account for any unmeasured of residual confounding which may have otherwise been explained the results we found.

2. - Have all participants received the same cessation intervention? This wasn't clear – and we know Varenicline / Bupropion / NRT play different roles in attenuating PCWG. Please provide information about their distribution between study groups.

As noted in response to the point above, there was no significant difference between trial arms in the use of pharmacotherapies. Pharmacotherapy use by trial arm has now been added to table 1.

3. - The final follow-up is only at 6 months (3 months after end of intervention). It is likely that any benefit of the weight management program will be dissipated already at 6-12 months after intervention. Ongoing intervention is obviously needed to sustain the observed beneficial effect. This aspect together with the very little interest in the weight management program by the large majority of smokers attending Stop Smoking Services should lead to a more in-depth discussion about their insignificant cost-effectiveness and lack of impact at population level. Needs to be addressed.

Added to the discussion:

The trial was also limited by a short-term follow-up at 6 months post quit (3 months post treatment intervention), therefore the long term consequences of the effect on weight gain prevention are unknown. However, meta-analysis data from randomised controlled trials suggests that 17% of weight lost is maintained 5 years after the end of moderate weight management interventions [32] so we might expect a similar trajectory here. This intervention in stop smoking services may therefore have limited impact in the long-term, nonetheless the short term increase

in diabetes risk associated with smoking cessation that is partly explained by weight gain [33] may be reduced, however this needs to be investigated in a larger trial, with a longer period of follow-up.

32. Anderson JW, Konz EC, Frederich RC, Wood CL. Long-term weight-loss maintenance: a meta-analysis of US studies. *The American journal of clinical nutrition*. 2001 Nov 1;74(5):579-84.

33. Pan A, Wang Y, Talaei M, Hu FB, Wu T. Relation of active, passive, and quitting smoking with incident type 2 diabetes: a systematic review and meta-analysis. *The lancet Diabetes & endocrinology*. 2015 Dec 1;3(12):958-67.

4. - High rate of loss to follow-up was also reported in the intervention arm of the study, in spite of the fact that the weight management program was offered for free. This needs to be addressed.

Added to the discussion:

Another limitation is the high rate of loss to follow-up in **both the control and treatment arms of the study. The provision of free weight management advice did not deter attrition. High attrition is common in smoking cessation trials and is usually due to participants returning to smoking** and as such the standard approach in analysis is to assume this [21].

5. - Study findings cannot be generalized as they are restricted only to a population of obese/overweight smokers. Add to limitation of the study,

Added:

Therefore, the generalisability of our findings are largely limited to these who are already overweight or obese.

6. - I acknowledge that prolonged abstinence is defined according to the widely used Russell Standard, but a good quality PCWG trial requires stronger measures of quantity and frequency of tobacco use and stronger compliance checks. In the specific case of eCO confirmed abstinence, the issue I have is that abstinence appears to be confirmed only at 12 and 26 weeks. Hard to establish continuous (vs intermittent) cessation with the approach described in this study. Cotinine assessments may be more useful – although if being prescribed an NRT as an intervention perhaps not possible.

We agree with this point however, NRT was a treatment option in this trial and so cotinine measurements were not appropriate.

7. - The observed mean difference of -1.9 Kg at 3 months (-1.3 Kg at 6 months) between study groups is clinically (as well as psychologically) irrelevant and specifically in a population of overweight and obese smokers (88% of the total). The practical implications of such slim change must be discussed.

If, the aim of this study was to address the obesity crisis we agree that our findings would be of limited importance. However, this was not the aim of our study; our aim was to prevent cessation-related weight gain and the results show that this was achieved (For example, at 3 months the observed mean difference is an actual weight loss of 1.3kg in the treatment group compared to 2kg weight gain in the control group – a difference to 3.3kg. Mean differences in table 3 are adjusted and so vary slightly) within the constraints and limitations we encountered. To clarify this we have added to the discussion:

The aim of the study was to prevent cessation related weight gain – which as discussed in the introduction has both clinical and psychological consequences, the most harmful of which may be considered as relapse to smoking. The trajectory of cessation related weight gain was seen by a consistent weight increase in the control group. In comparison, the treatment group lost weight at each of these time points.

Reviewer: 3

Reviewer Name

Steve Sutton

Institution and Country

Moffitt Cancer Center and Research Institute
Tampa, FL USA

Please state any competing interests or state 'None declared':
'None declared'

Please leave your comments for the authors below

The manuscript presents results for a study assessing the effectiveness of weight management treatment as a supplement to smoking cessation treatment among treatment seekers. The topic is important and the novel approach to incorporating weight management treatment is to be applauded. Unfortunately, slow enrollment eventuated in stopping the study short of the target sample size. The following are my primary concerns regarding the method.

1) The final sample is less than what was intended. It would help to provide the planned sample size and the analysis that was used to generate that target. This may be provided in the referenced protocol paper, but needs to be presented here for the reader to understand how much less the actual sample size was relative to the planned sample size.

Added to methods:

Our original sample size had been conservative assuming we would see 20% weight gain in the treatment group, rather than the weight loss we actually saw. This would have required 32 quitters in each trial arm at 12 weeks; instead, we achieved 8-10 quitters at that time point. To achieve 32 quitters in each trial arm, based on a conservative estimate that only 20% of those recruited would be quit at 12 weeks, we wanted to recruit 160 per trial arm. Instead, we only recruited 76 smokers in total, but the quit rates in the treatment group were higher than expected.

2) A related issue is the decision to stop recruitment and end the study early. The factors that determined this decision should be described in the Introduction. This decision cannot be based on a posthoc power analysis, or justified by a posthoc power analysis. With such a small sample size for the analysis of the primary outcome, there is a greater probability that the observed group difference occurred by chance. Deciding to stop at a point in the study when the main outcome showed a significant difference may be taking advantage of chance factors.

The decision to stop the trial was not based on post-hoc power analysis. This power analysis was only done after analysis of our outcomes. The trial was stopped because it was not economical to continue to recruit and funding had run out.

We have added to the methods:

Difficulty in recruiting resulted in it being economically unaffordable to continue the trial; therefore, the decision was made to stop the trial early. Following the full analysis of our outcome data we conducted reverse power calculations. This showed that we had >80% power, at an alpha...

3) Given the small sample sizes and the presented values in Table 1, I presume that there were no statistically significant group differences in the characteristics listed. It would be helpful to formally present that information in text or in the table. For any group difference with $p < .10$ (maybe higher),

that characteristic should be included in any analysis focusing on treatment effects. On page 12, there is a statement that there are no treatment group differences. However, the statement directs the reader to Table 2, which presents data on 'participants' versus 'decliners'. And the statement follows a paragraph describing reasons for declining to participate. Both sets of group comparisons should be presented.

Table 1 and 2 were numbered the wrong way round and this has now been corrected.

As recruitment did not reach sample size we understand we cannot assume effective randomisation. We have now investigated whether there were any significant differences in baseline characteristics (see footnotes on Table 1) between the intervention and control groups. We found a significant difference ($p < 0.1$) only for baseline BMI ($p = 0.82$) and age ($p = 0.53$) and have reported on this in the text:

[Baseline characteristics of participants were investigated between the similar in the control and treatment groups as failure to recruit to target sample size could have resulted in an imbalance.

Those in the control group were significantly younger and with a higher BMI than the treatment group (Table 2) and adjustment was made for this in the analysis of outcomes.]

We have also added into table 4 the adjusted results for all the outcomes (adjusted for BMI and age).

4) The analysis of the primary outcome used ANCOVA. Given the small sample size and the lack of direct randomization of the data analytic treatment groups, there should be an attempt to control for baseline characteristics that may differ by analytic group and/or are associated with the outcome measure. Even marginally statistically significant differences and associations are less likely to be observed given the small sample size. Therefore, the approach may have to be conceptual. That is, the analysis would incorporate any conceptually relevant baseline characteristic, especially one that shows any sign of a difference between groups.

We have added to the methods:

...as this comparison was made in abstainers only, and not those randomised to control or treatment groups, investigation of baseline differences was conducted between these groups. To identify likely confounding variables selected baseline characteristics of age, weight, BMI, FTND,

gender, ethnicity and pharmacotherapy were compared, between quitters at 4, 12 and 26 weeks post quit in the control and treatment groups, using t-tests and chi-squared statistics.

We have added to the results:

No confounding variables were identified in the quitters between treatment and control groups at week 4 and 12 post-quit (results no shown), age was a potential confounding variable for those who remained quit at 26 weeks and adjustment for age was included in the analysis of difference in weight between the trial arms.

5) Given the small sample size, conclusions based on non-significant differences should not be highlighted (e.g., hunger ratings in the abstract).

The reference to hunger ratings in the abstract has been removed.

The following are relatively minor concerns.

1) There are numerous baseline measures presented in Table 1 where the number of missing observations for the Treatment group (4-7 of 37) is much greater than for the Control group (1-2 of 39). It would help to have an explanation for the degree of missing baseline data.

We have added to the limitations:

Additionally, not all data was fully completed at baseline, there were disproportionately more missing values in the treatment than the control arm but we have no reason why this might be the case.

2) A second minor concern with Table 1 is the mean and standard deviation for FTND in the Treatment group. These values do not seem reasonable.

We have rechecked these values and found a typo in the individual level data, the values for FTND have now been updated:

Control: 5.7 (2.2) Treatment: 5.8 (2.1)

3) There appears to be a typographical error in Table 4. For 12 weeks and 26 weeks, the column titles for group difference in change has '%' written where I believe 'OR' was intended.

This has been corrected

4) As I understand it, the 4-week and 26-week measures are anchored by the date of the baseline measure and the 12-week measure is anchored by quit. If there are different anchors, it would help to have descriptive statistics on the quit date relative to baseline. Table 3, though, list all follow-ups

as anchored by quit date.

All measures are consistently compared to values at baseline, tables 3 and 4 describe change in outcome from baseline (top line, first column). The times at which the values are measured are 4, 12 and 26 weeks after quit date, this stated in table 3 but to avoid confusion has now been clarified in table 4.

VERSION 2 – REVIEW

REVIEWER	Francesco Pistelli Pulmonary Unit Cardiothoracic and Vascular Department Pisa University Hospital Italy
REVIEW RETURNED	02-Nov-2019
GENERAL COMMENTS	The authors have adequately reviewed the paper according to the editorial requests and reviewers' comments and it has now improved. This reviewer has no more comments.
REVIEWER	Polosa Riccardo University of Catania
REVIEW RETURNED	25-Oct-2019
GENERAL COMMENTS	Point 3. <<This intervention in stop smoking services may therefore have limited impact in the long-term, nonetheless the short term increase in diabetes risk associated with smoking cessation that is partly explained by weight gain [33] may be reduced, however this needs to be investigated in a larger trial, with a longer period of follow-up.>> As acknowledged by the authors, the new statement above is highly speculative (particularly when considering the trivial reduction in weight gain) and should be deleted. I suggest: <<This intervention in stop smoking services may therefore have limited impact>>. Point 7. I perfectly understand the aim of the study. It is a fact that this trial shows that reduction in weight gain is clinically as well as psychologically insignificant and more so in overweight and obese smokers. The practical implications of such slim change for the obese smoker (and not for the obese population) would add value to the discussion and should be addressed.
REVIEWER	Steve Sutton Moffitt Cancer Center & Research Institute Tampa, FL, USA
REVIEW RETURNED	15-Nov-2019
GENERAL COMMENTS	The authors adequately responded to the reviewer comments of the previous version. The revised manuscript is much-improved. I have no additional comments.

VERSION 2 – AUTHOR RESPONSE

The remaining reviewer comments are detailed below with the changes made in response

Point 3. <>

As acknowledged by the authors, the new statement above is highly speculative (particularly when considering the trivial reduction in weight gain) and should be deleted. I suggest: <>.

Authors response to point 3: Changes have been made as suggested above.

Point 7. I perfectly understand the aim of the study. It is a fact that this trial shows that reduction in weight gain is clinically as well as psychologically insignificant and more so in overweight and obese smokers. The practical implications of such slim change for the obese smoker (and not for the obese population) would add value to the discussion and should be addressed.

Authors response to point 7, the following has been added:

It is also important to note that the magnitude of weight change is relatively small (mean reduction 1.1kg at 12 weeks), only 1.3% weight loss considering the mean weight of all participants at baseline was 86.9kg.